# Research on Ultra-Wideband NLFM Waveform Synthesis and Grating Lobe Suppression

**DOI:** 10.3390/s22249829

**Published:** 2022-12-14

**Authors:** Shuyi Liu, Yan Jia, Yongqing Liu, Xiangkun Zhang

**Affiliations:** 1Key Lab of Microwave Remote Sensing, National Space Science Center, Chinese Academy of Sciences, Beijing 100190, China; 2School of Electronic, Electrical and Communication Engineering, University of Chinese Academy of Sciences, Beijing 100049, China

**Keywords:** SWW, NLFM, UWB, Fresnel ripple, random frequency hopping

## Abstract

Ultra-wideband (UWB) nonlinear frequency modulation (NLFM) waveforms have the advantages of low sidelobes and high resolution. By extending the frequency domain wideband synthesis method to the NLFM waveform, the synthetic bandwidth will be limited, and the grating lobe will grow as the number of subpulses increases at a fixed synthetic bandwidth. Aiming for the highly periodic grating lobes caused by equally spaced splicing and small subpulse time-bandwidth products (TxBW), a multisubpulse UWB NLFM waveform synthesis method is proposed in this paper. Random frequency hopping and spectral correction are utilized to disperse the energy of periodic grating lobes and optimize the matched filter of the subpulse, thereby reducing notches and Fresnel ripples in the synthesized spectrum. The results of the hardware-in-the-loop simulation experiment show that the peak sidelobe ratio (PSLR) and the integral sidelobe ratio (ISLR) of the NLFM synthetic wideband waveform (SWW) obtained by 50 subpulses with a bandwidth of 36 MHz are improved by 4.8 dBs and 4.5 dBs, respectively, when compared to the frequency domain wideband synthesis method, and that the grating lobe is suppressed by an average of 10.6 dBs. It also performs well in terms of point target resolution, and it has potential for 2D radar super-resolution imaging.

## 1. Introduction

With the advancement of radar imaging and the spread of microwave remote sensing equipment to such applications as terrain exploration and electronic reconnaissance, both the military and civilian industries have increased their resolution requirements for radar systems. Obtaining extensive and refined information about targets via UWB waveforms is a crucial direction of development for radar imaging. Typically, radars with a range resolution of 0.1 m require a system operation bandwidth greater than 1.5 GHz. Even if the IQ demodulation receiver is used to split the echo into two channels, the sampling rate of each channel after digitization is still larger than 1.5 Gsps, which places a tremendous amount of stress on the analogue-to-digital (AD) device, instantaneous data transfer, and storage, and it increases the cost of the system. Wideband synthesis technology disassembles the large bandwidth waveform into several stepped-frequency (SF) subpulses, which are spliced at the receiver to obtain SWWs for the high-range resolution profile (HRRP). Since the bandwidth of each subpulse is relatively small, it can efficiently reduce the requirements for an instantaneous bandwidth and sampling rate.

The SF frequency modulation (FM) subpulse with uniform carrier frequency spacing has severe periodic range grating lobes, which will produce fake targets and raise the false-alarm rate. The grating-lobe suppression methods for SWW can be classified into two categories: signal processing and parameter design. The signal processing method primarily suppresses grating lobes by adding a linear or nonlinear window to the synthesized spectrum. In Reference [1], the subpulse and the synthesized spectrum are separately windowed to suppress the grating lobes at the cost of resolution. In Reference [2], the apodization filter is applied to SWW to suppress grating lobes and sidelobes without resolution loss. However, the suppression effect varies depending on the system parameter, and the calculation is substantial, making its implementation in a real-time radar-processing system challenging. The parameter design method makes it difficult to generate grating lobes by designing appropriate SF FM waveform parameters. Reference [3] analyzes the relationships between subpulse bandwidth, pulse duration, carrier frequency spacing, and grating lobes. Through subpulse parameter design, the grating lobe is located at the zero point of the range-compressed SWW, but usually only part of the grating lobes can be suppressed, and sophisticated parameter optimization will also make system design more complex. Reference [4] proposes a SF linear frequency modulation (LFM) waveform with nonuniform carrier frequency spacing to break the periodicity of the grating lobe, achieving effective grating-lobe suppression. By developing an accurate and approximative model of the LFM subpulse spectrum with a small TxBW, Reference [5] proposes a low blind-distance ultra-wideband waveform design method based on a spectral fluctuation period and Fresnel integral windowing that reduces the number of grating lobes by over 50% and suppresses the highest grating-lobe level by at least 4 dBs. However, subpulse spectra in NLFM SWW are distinct, and the derivation of the Fresnel integral is complex, making it challenging to apply to NLFM SWW. Reference [6] determines that intrapulse and interpulse amplitude and phase errors introduced by device nonideality are the cause of grating lobes and proposes using strong scattering point echoes for amplitude and phase error compensation, as well as providing a grating-lobe suppression method for LFM SWW from a systems application perspective.

In the aforementioned studies on SF FM radar and its grating-lobe suppression, the majority of subpulse waveforms adopt LFM waveforms, and there are few studies on the synthesis of NLFM waveforms. Compared with the LFM waveform, the NLFM waveform may obtain a low ratio between the main- and sidelobes without weighting, and it has the benefits of low interception and small waveform loss [7]. Therefore, research on UWB NLFM SWW design and grating-lobe suppression based on wideband synthesis technology is conducive to obtaining super-resolution, high-quality radar images with low sidelobes. It is of great significance to the monitoring of weak targets against a background of strong scattering targets and clutter, and it offers advantages in the fields of meteorological monitoring and weak target identification. Our research on UWB NLFM waveform synthesis is based on the frequency domain wideband synthesis method of the LFM waveform. The main research contents consist of: (1) building a SF FM subpulse model for UWB NLFM waveform synthesis and deriving the generation of subpulses; (2) applying the traditional frequency domain wideband synthesis method to UWB NLFM waveform synthesis, and analyzing the reasons for range grating lobes according to the simulation results; (3) combining random frequency hopping and spectrum correction technology, proposing a multisubpulse UWB NLFM waveform synthesis method to suppress periodic grating lobes caused by splicing at equal intervals and subpulses with small TxBW; (4) building a hardware-in-the-loop simulation system for SF FM radar using general microwave instruments, and conducting ideal single-point and multipoint target simulation experiments to demonstrate the effectiveness of the proposed method.

## 2. Wideband Synthesis of LFM Waveform

### Principle of Frequency Domain Wideband Synthesis

Assume the number of LFM subpulses is N, the bandwidth is B, and the carrier frequency step size is Δf. The subpulse time domain signal is [8]
$$\begin{array}{l} p_{i}\left(t\right)=A_{i}\cdot rect\left(\frac{t-iT_{r}}{T_{p}}\right)\cdot \exp\left(j2\pi f_{i}t\right)\cdot \exp\left(j\pi Kt^{2}\right)\\ \;\;=A_{i}\cdot u\left(t-iT_{r}\right)\cdot \exp\left(j2\pi f_{i}t\right) \end{array}$$
(1)$$f_{i}=f_{0}+\left(i-1\right)\Delta f\;i=0,1\ldots N-1$$
where $T_{p}$ denotes pulse width, $T_{r}$ denotes pulse repetition interval (PRI), $K=B/T_{p}$ is the chirp rate, $f_{0}$ is the center frequency of the first transmitted subpulse, $A_{i}$ denotes the ith subpulse amplitude, and $u\left(t\right)=rect\left(\frac{t}{T_{p}}\right)\exp(j\pi Kt^{2})$ denotes the baseband LFM waveform.

After demodulation, the baseband subpulse echo is
(2)$$n_{i}\left(t\right)=B_{i}\cdot rect\left(\frac{t-iT_{r}-\tau }{T_{p}}\right)\cdot \exp\left(-j2\pi f_{i}\tau \right)\cdot u\left(t-iT_{r}-\tau \right)$$
where $B_{i}$ is the ith baseband subpulse amplitude, τ is the echo delay of the subpulse, and c denotes the light speed.

Matched filtering, up-sampling, spectrum displacement, and subpulse splicing are performed on each baseband subpulse [9], as shown in Figure 1, and the synthesized spectrum is [10]
$$\begin{array}{l} S_{syn}\left(f\right)=\overset{N}{\underset{i=1}{\sum }}C_{i}rect\left(\frac{f_{r}-\delta f_{i}}{B}\right)\exp\left(-j2\pi \left(f_{r}-\delta f_{i}\right)\tau \right)\exp\left(-j2\pi f_{i}\tau \right)\\ \;\;\;=C\cdot rect\left(\frac{f_{r}}{B_{syn}}\right)\cdot \exp\left(-j2\pi f_{r}\tau \right)\cdot \exp\left(-j2\pi f_{c}\tau \right) \end{array}\;$$
(3)$$\delta f_{i}=f_{i}-f_{c}\;B_{syn}=B+\left(N-1\right)\Delta f$$
where C denotes the amplitude of the synthesized wideband spectrum, $B_{syn}$ is the synthetic bandwidth of the SWW, $f_{c}$ is the center frequency of the SWW, and $\delta f_{i}$ is the frequency shift amount.

Performing the inverse fast Fourier transform (IFFT) on Equation (3), the range-compressed SWW is
(4)$$s_{syn}\left(t\right)=D\cdot \sin c\left(B_{syn}\left(t-\tau \right)\right)\cdot \exp(-j2\pi f_{c}\tau )$$
where D is the amplitude of the range-compressed SWW.

The range resolutions of the single subpulse and SWW are shown in Equation (5). After wideband synthesis, the range resolution of the SF LFM waveform is improved.
(5)$$\rho =\frac{c}{2B}\rho_{syn}=\frac{c}{2B_{syn}}=\frac{c}{2\left(B+\left(N-1\right)\Delta f\right)}$$

## 3. Wideband Synthesis of NLFM Waveform

### 3.1. Stepped-Frequency FM Waveform Model

Generally, the principle of stationary phase (POSP) is used to generate NLFM waveforms based on the classical window function. Suppose the NLFM waveform can be represented in the time domain and frequency domain as [11]
(6)s(t)=a(t)exp(jφ(t))
(7)U(f)=A(f)exp(jϕ(f))
where a(t) is the instantaneous amplitude, typically a(t)=1, φ(t) is the instantaneous phase function, A(f) is the amplitude spectrum, and ϕ(f) is the phase spectrum of the NLFM waveform.

The output of the corresponding matched filter is
(8)$$Y\left(f\right)=U\left(f\right)\cdot U^{*}\left(f\right)=A^{2}\left(f\right)=W\left(f\right)$$
where W(f) represents the power spectral density (PSD) of the NLFM waveform, which is typically a window function such as Hamming, Kaiser, Taylor, etc. Since the autocorrelation function corresponds to PSD, the NLFM waveform can be attributed to designing the waveform according to the expected PSD, such that the amplitude spectrum satisfies $A^{2}\left(f\right)=W\left(f\right)$ [12].

According to POSP, calculate the group delay T(f) and its inverse function [13], and the time-frequency function f(t) of the NLFM waveform can be obtained as
$$T\left(f\right)=K_{1}\int_{-\infty }^{f}W\left(x\right)dx-B_{syn}/2\leq f\leq B_{syn}/2\;T\left(B/2\right)=K_{1}\int_{-B_{syn}/2}^{B_{syn}/2}W\left(x\right)dx=T_{syn}$$
(9)$$f\left(t\right)=T^{-1}\left(f\right)$$
where $K_{1}$ is a constant defined by the group delay boundary condition, and $T_{syn}$ is the duration of the NLFM waveform.

Using the time-frequency function f(t), determine the phase function φ(t), and then substitute it into Equation (6) to obtain the NLFM waveform.
(10)$$\varphi \left(t\right)=2\pi \int_{-\infty }^{t}f\left(x\right)dx$$

Determine the instantaneous phase of each baseband FM subpulse by dividing the time-frequency function f(t) by the subpulse bandwidth B and the carrier frequency spacing Δf. Each baseband subpulse is modulated to $f_{i}\left(i=0,1\ldots N-1\right)$, respectively, to generate FM subpulse transmission waveforms, as shown in Figure 2 and Equation (11).
(11)$$\begin{array}{l} p_{i}\left(t\right)=A_{i}rect\left(\frac{t-iT_{r}}{T_{p_{i}}}\right)\cdot \exp\left(j2\pi f_{i}t\right)\cdot \exp\left(j\phi_{i}\left(t-iT_{r}\right)\right)\\ \;\;=A_{i}\omega_{i}\left(t-iT_{r}\right)\cdot \exp\left(j2\pi f_{i}t\right) \end{array}$$

### 3.2. Frequency Domain Wideband Synthesis of NLFM Waveform

According to the subpulse echo, the transfer function of each subpulse matched filter is
(12)$$H_{i}\left(f\right)=FFT\left[w_{i}\left(t\right)\right]$$

Then, remove the overlapping spectrum and perform subpulse splicing to obtain the synthesized NLFM spectrum, as shown in Figure 3 and Equation (13).
(13)$$S_{syn}\left(f\right)=\sum_{i=0}^{N-1}S_{i}\left(f-\delta f_{i}\right)\cdot rect\left(\frac{f-\delta f_{i}}{B}\right)$$
where $S_{i}\left(f\right)$ is the output of subpulse matched filter. Perform IFFT on Equation (13) to obtain a range-compressed NLFM SWW with low sidelobes, whose synthetic bandwidth is the same as Equation (5). In addition, the equivalent pulse width is
(14)$$T_{syn}\approx \frac{\Delta f}{B}\sum_{i=0}^{N-1}T_{p_{i}}+\frac{B-\Delta f}{B}T_{p_{1}}$$
which is improved as compared to the single pulse [14].

### 3.3. Cause of Grating Lobes

NLFM SWW reduces system bandwidth and improves range resolution at the expense of high grating lobes. Under the uniform carrier frequency spacing, the synthesized spectrum is discontinuous with Δf as the period due to subpulse splicing, and it appears as pulse-like periodic grating lobes with an interval of c/2Δf in the range. The grating lobe grows as the number of subpulses rises, which has a significant impact on target-detection accuracy.

#### Random Carrier Frequency Spacing

Reducing the carrier frequency spacing (or raising the spectral overlap rate) can efficiently suppress grating lobes, but more subpulses are required to generate a SWW with a fixed synthetic bandwidth. In fact, an excessive number of subpulses reduces the time correlation between subpulses, which complicates subpulse motion compensation. Hence, the carrier frequency spacing should not be too small.

Previous research on LFM SWW has demonstrated that subpulses are spliced with nonuniform spacing, breaking the periodicity of the range grating lobes [4,15]. With the advancement of the field-programmable gate array (FPGA) and digital signal processing, random frequency hopping and arbitrary waveform generation can be easily implemented in software-defined radar (SDR) [16]. This section will explore the effect of random frequency hopping on grating-lobe suppression in NLFM SWW, where subpulses are nonuniformly spliced via random carrier frequency spacing to reduce the periodicity of the discontinuity in the synthesized spectrum and to disperse the energy in the grating lobe [17].

Based on SF FM subpulses with uniform carrier frequency spacing, we define a nonuniform step size, $\Delta f_{n}$
(15)$$\Delta f_{n}=\Delta f+U_{n}U_{n}\in \left[-\frac{1}{2}\Delta f_{overlap},\frac{1}{2}\Delta f_{overlap}\right]$$
where $\Delta f_{overlap}$ represents the overlapping spectrum at uniform carrier frequency spacing, and $U_{n}$ is a uniformly distributed random variable. To avoid grating lobes caused by notches in the synthesized spectrum, $U_{n}$ should not exceed half of the overlapping spectrum.

Figure 4 shows the range-compressed NLFM SWW synthesized by eight subpulses based on the Hamming window. There are periodic grating lobes in the range at uniform carrier frequency spacing; the difference between the time-bandwidth products of the subpulses is small, and the subpulse spectra on both sides are less impacted by the time-bandwidth products [18]. At this time, grating lobes are mainly caused by equally spaced splicing, so most of the grating lobes can be suppressed at nonuniform carrier frequency spacing. When the synthetic bandwidth is fixed and the number of subpulses is increased to 20, notches occur in the synthesized spectrum, the in-band Fresnel ripple increases, and the periodic grating lobe grows substantially. The random carrier frequency spacing can disperse the periodic grating-lobe energy, but the grating-lobe level is high overall, weakening the grating-lobe suppression effect, as shown in Figure 5c,d.

Figure 6 demonstrates how the TxBW of each subpulse changes in the NLFM SWW. When the synthetic bandwidth is fixed, the subpulse TxBW decreases significantly with the increase of the subpulse, the Fresnel ripple gradually grows from the middle to the two sides [19], the effective energy in the bandwidth decreases [20], and the overlapping spectrum of adjacent subpulses also shrinks [21]. Random carrier frequency spacing can break the periodicity of grating lobes, but it hardly improves the notch and in-band ripple caused by subpulse spectral edge degradation and overlapping spectrum shrinkage. Therefore, the grating-lobe suppression effect of nonuniform splicing gradually weakens with the increase of subpulses in NLFM SWW.

### 3.4. Multisubpulse UWB NLFM Waveform Synthesis Method

The grating lobe of NLFM SWW is closely related to notches and in-band Fresnel ripples, which makes the suppression effect of random carrier frequency spacing weaken gradually as the number of subpulses increases, thereby restricting its application in the case of a large number of subpulses. In this paper, a multisubpulse UWB NLFM waveform synthesis method based on the principle of frequency domain wideband synthesis is proposed. The main idea is to combine the advantages of random frequency hopping and spectral correction technology, introducing nonuniformity to disperse periodic grating-lobe energy via random carrier frequency and optimizing the matched filter of each subpulse in sections based on spectral correction technology. By suppressing the Fresnel ripple and edge jump of the subpulse spectrum, the notch and in-band ripple in the synthesized spectrum are minimized, approximating an ideal window function and effectively suppressing grating lobes.

For an ideal NLFM waveform, the transfer function of the spectral correction filter is
(16)$$H\left(f\right)=\frac{W\left(f\right)}{{\left|U\left(f\right)\right|}^{2}}U^{*}\left(f\right)-B_{syn}/2\leq f\leq B_{syn}/2$$

In NLFM SWW, each subpulse matched filter is optimized according to the PSD of the ideal NLFM waveform, and the subpulse spectrum edge jump and in-band ripple are reduced by suppressing the in-band fluctuation of the matched filter. Let each baseband subpulse spectrum be $Y_{i}\left(f\right)$. For each subpulse, the transfer function of the spectral correction filter is
$$H_{i}^{'}\left(f\right)=\frac{W\left(f+\delta f_{i}\right)}{{\left|Y_{i}\left(f\right)\right|}^{2}}\cdot Y_{i}^{*}\left(f\right)\cdot rect\left(\frac{f}{B}\right)$$
(17)−B/2≤f≤B/2

Up-sampling, spectrum displacement, and subpulse splicing are performed on the output of the subpulse spectral correction filter, respectively, and the synthesized NLFM spectrum is shown in Equation (18) and Figure 7.
(18)$$\begin{array}{l} S_{syn}\left(f\right)=\overset{N-1}{\underset{i=0}{\sum }}Y_{i}\left(f-\delta f_{i}\right)\cdot H_{i}^{'}\left(f-\delta f_{i}\right)\\ \;\;\;=\overset{N-1}{\underset{i=0}{\sum }}S_{i}^{'}\left(f-\delta f_{i}\right)\cdot rect\left(\frac{f-\delta f_{i}}{B}\right) \end{array}$$

Figure 8 and Table 1 show the comparative simulation results of Figure 5 with the same parameters after spectral correction. After optimizing the subpulse matched filter with spectral correction, the notch and discontinuity caused by the subpulses with small TxBW in the synthesized spectrum are reduced. Compared with Figure 5c, by suppressing the in-band ripple and spectral jump, the grating lobe is lowered by an average of 4.76 dBs, while the ISLR and PSLR are also reduced. However, due to equally spaced splicing, periodic grating lobes still exist in range-compressed NLFM SWW. Under random carrier frequency spacing, the nonuniform splicing of the subpulses suppresses grating lobes by about 4.91 dBs on average, as compared with uniform splicing, and the energy of the periodic grating lobe is dispersed, with the near-end grating lobe being suppressed more obviously, as shown in Figure 8c. In conclusion, the multisubpulse UWB NLFM waveform synthesis method can disperse the periodic grating-lobe energy and suppress the grating lobe by reducing in-band ripples and notches of the synthesized spectrum, breaking the periodicity of the discontinuity. Both the PSLR and ISLR of SWW are superior to those of the traditional wideband synthesis method and uniform splicing, proving the effectiveness of the proposed method.

## 4. Experimental Verification and Analysis

### 4.1. Experimental Verification and Analysis

To verify the effectiveness of the multisubpulse UWB NLFM waveform synthesis method and the feasibility of engineering application, a hardware-in-the-loop simulation system for SF FM radar is built using general microwave instruments [22]. It consists of a computer, a router, an oscilloscope from Rohde & Schwarz, a vector signal generator, and a spectrum analyzer from Ceyear. The connection block diagram and physical objects are shown in Figure 9 and Figure 10. In addition, the system constructs an internal calibration path by directly coupling the transmitted waveform to the receiver via the delay line in order to eliminate the interpulse system distortion, delay error, and phase jump at the splicing point [23]. The transmitter and receiver are directly connected by one or more coaxial cables to simulate a radar system observing one or more ideal point targets in order to verify the performance of the NLFM SWW and the resolution of the point targets. Table 2 demonstrates the experimental parameter settings.

### 4.2. Single-Point Target Experiment

SF FM subpulses are transmitted in accordance with the experimental parameters indicated in Table 2, and the ideal single-point target is simulated by the internal calibration signal in the system.

As depicted in Figure 11, the traditional frequency domain wideband synthesis method is extended to the synthesis of the UWB NLFM waveform, and severe periodic grating lobes at equal carrier frequency spacing are observed. When the synthetic bandwidth is fixed, the subpulse TxBW decreases significantly from the middle to both sides as the subpulse increases, increasing subpulse Fresnel ripples and inherently introducing notches into the synthesized spectrum.

Based on the frequency domain wideband synthesis method, the multisubpulse UWB NLFM waveform synthesis method introduces nonuniformity via random carrier frequency spacing to disperse energy in periodic grating lobes and improves the subpulse matched filter via spectral correction to reduce notches and Fresnel ripples in the synthesized spectrum. Therefore, the grating lobe of UWB NLFM SWW can be effectively suppressed, improving the accuracy of target identification.

Figure 12 and Table 3 show the results of the range-compressed UWB NLFM SWW by the proposed method. In accordance with Figure 7, each subpulse is processed by pulse compression, spectrum displacement, and nonuniform subpulse splicing, and the bandwidth and time resolution of the SWW are 1.506 GHz and 0.8667 ns, respectively. The PSLR and ISLR both surpass the frequency domain wideband synthesis method by 4.8 dBs and 4.5 dBs, respectively, and the overall level of the grating lobe is suppressed by about 10.6 dBs, demonstrating the proposed method’s ability to suppress grating lobes for UWB NLFM SWW.

### 4.3. Single-Point Target Experiment

In order to verify the point target resolution performance of UWB NLFM SWW, two-point and three-point target experiments are carried out through coaxial cables. The group delay of the coaxial cables used in the two experiments is shown in Table 4.

The experimental results of the two-point and three-point targets are shown in Figure 13 and Figure 14, where the peak at time zero is the internal calibration signal of the system. The time resolution of a single subpulse is about 26.6 ns, and the minimum delay difference between targets in the two-point and three-point experiments is 8.18 ns and 3.607 ns, respectively. Both of these are less than the time resolution of a single pulse, so a single pulse cannot tell them apart. After processing with the proposed method, the time resolution rises to 0.8667 ns, and targets with delay differences smaller than the time resolution of a single subpulse can be distinguished, demonstrating the point target resolution of the UWB NLFM SWW.

## 5. Conclusions

Wideband synthesis technology is an efficient approach to achieving high range resolution, which can reduce the requirements for sampling rate and instantaneous bandwidth. NLFM SWW will contain periodic grating lobes due to equally spaced splicing when the frequency domain wideband synthesis method of the LFM waveform is applied to NLFM waveform synthesis. The subpulse TxBW decreases significantly with increasing subpulses at a fixed synthetic bandwidth, resulting in notches and intensified Fresnel ripples in the synthesized spectrum, and the grating lobe of the NLFM SWW increases in range. In response, this paper proposes a multisubpulse UWB NLFM waveform synthesis method that utilizes random carrier frequency spacing and spectral correction technology to disperse grating-lobe energy and improve subpulse matched filters to reduce notches and ripples in the synthesized spectrum, hence significantly suppressing the SWW grating lobe. To verify the effectiveness of the proposed method, a hardware-in-the-loop simulation system for SF FM radar is built to analyze NLFM SWW performance. The experimental results for single-point, two-point, and three-point targets show that, after processing 50 subpulses with a bandwidth of 36 MHz by the proposed method, the time resolution of range-compressed NLFM SWW is improved to 0.8667 ns, which is approximately 30.69 times higher than that of a single subpulse. The PSLR and ISLR are improved by 4.8 dBs and 4.5 dBs, respectively, compared to the frequency domain wideband synthesis method, and the grating lobe is reduced by an average of 10.6 dBs. In addition, it has excellent point target resolution and can distinguish targets that cannot be distinguished with a single subpulse. Based on the results in the paper, research on 2D imaging algorithms and system designs suitable for SWW can be carried out in the future to enhance the resolution and imaging quality of SF FM radar [24].

## Figures and Tables

**Figure 1 sensors-22-09829-f001:**
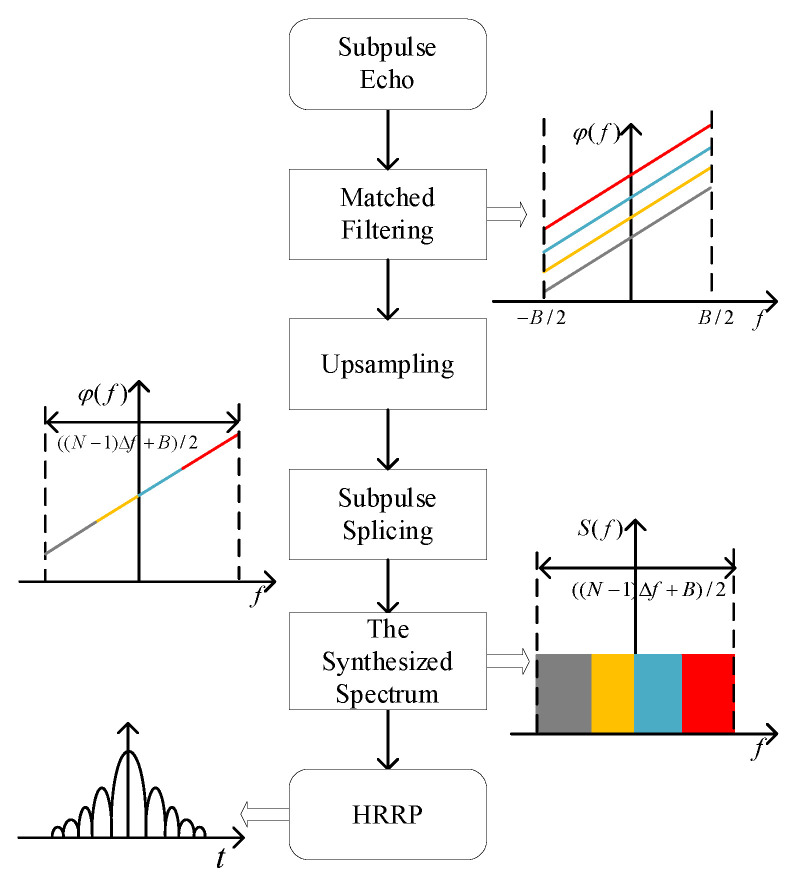
Frequency domain wideband synthesis of LFM waveform.

**Figure 2 sensors-22-09829-f002:**
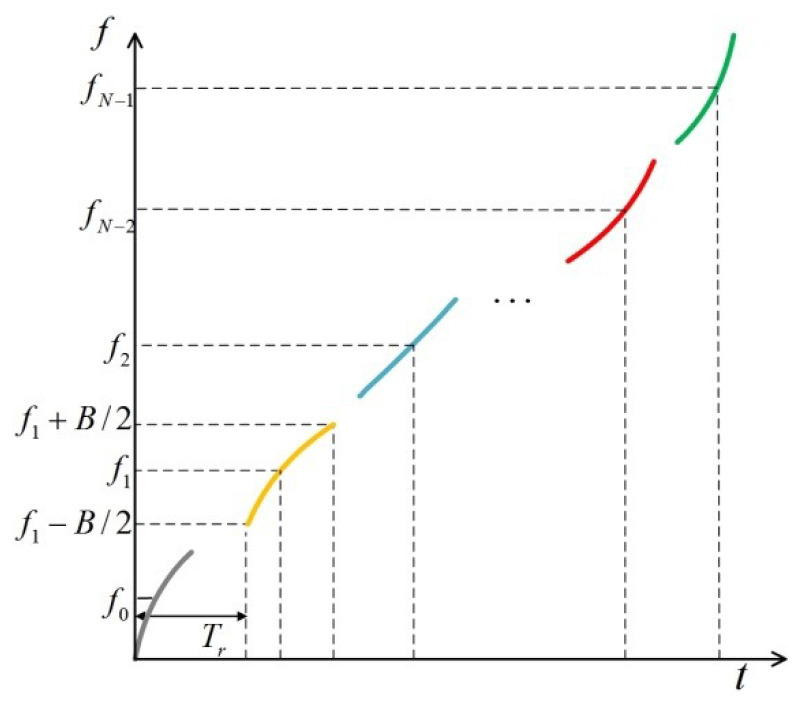
SF FM subpulse Model.

**Figure 3 sensors-22-09829-f003:**
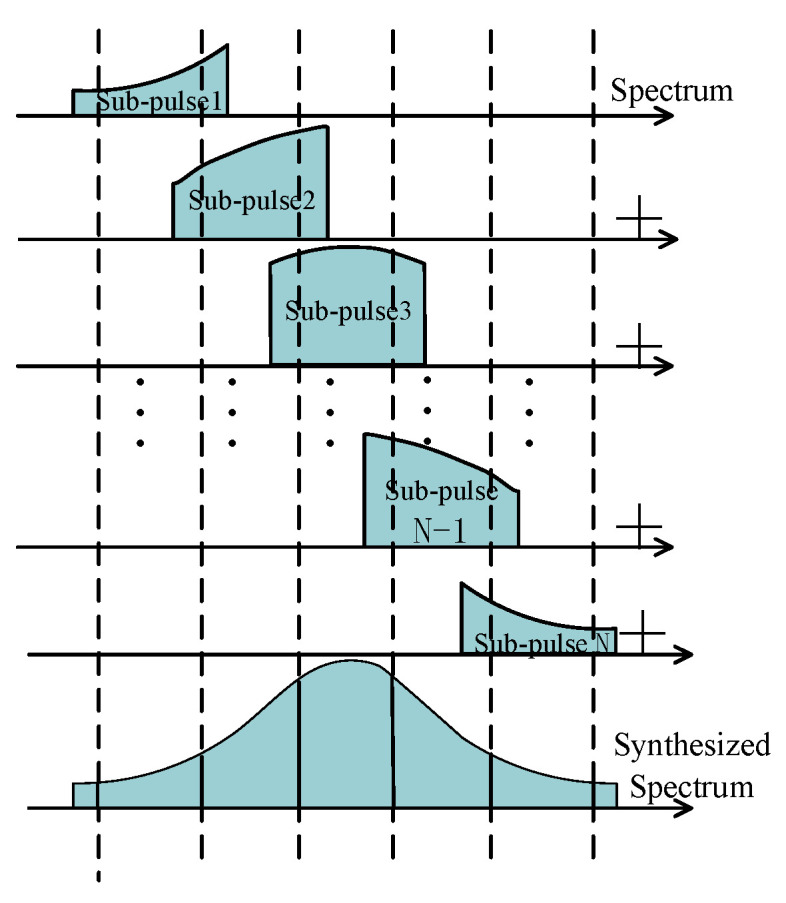
SF FM subpulse splicing.

**Figure 4 sensors-22-09829-f004:**
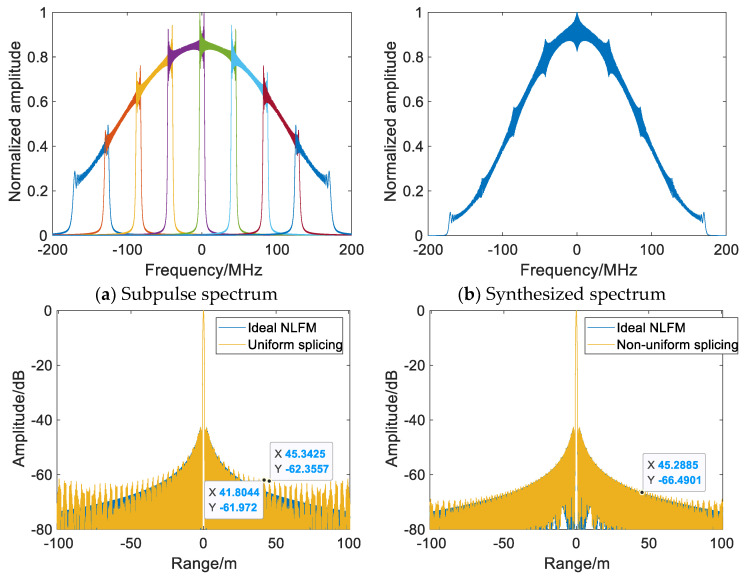
Range-compressed NLFM SWW ($N=8\;B=50\;\mathrm{MHz}\;\Delta f=42.5\;\mathrm{MHz}\;T_{syn}=145\;µs$).

**Figure 5 sensors-22-09829-f005:**
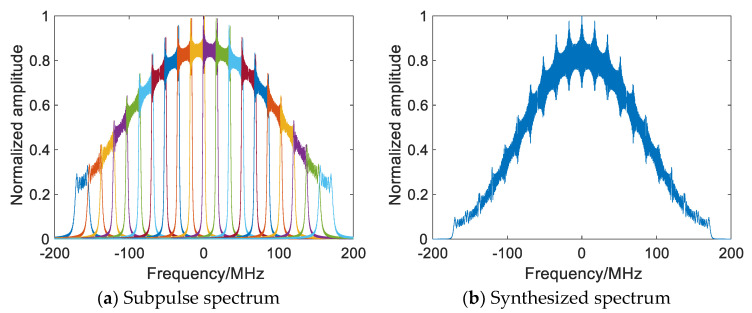
Range-compressed NLFM SWW (N=20 B=20.7 MHz Δf=17.2 MHz T=145 µs).

**Figure 6 sensors-22-09829-f006:**
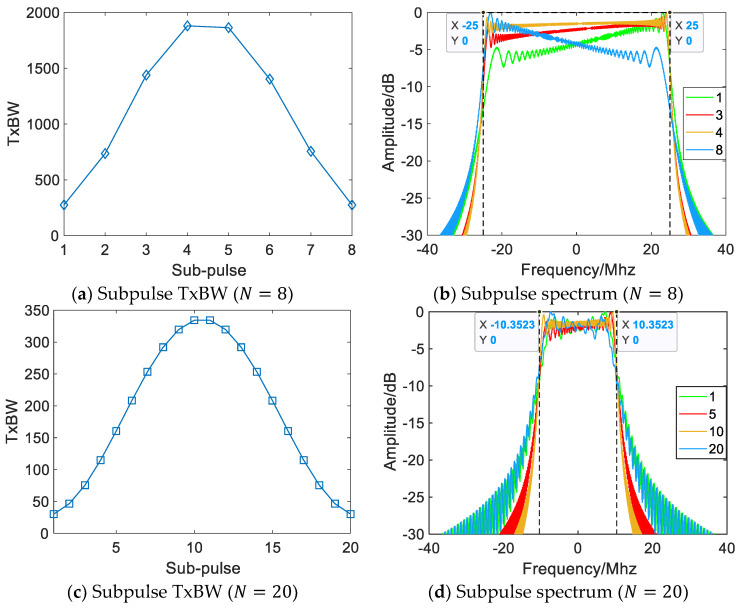
Subpulses at different TxBWs.

**Figure 7 sensors-22-09829-f007:**
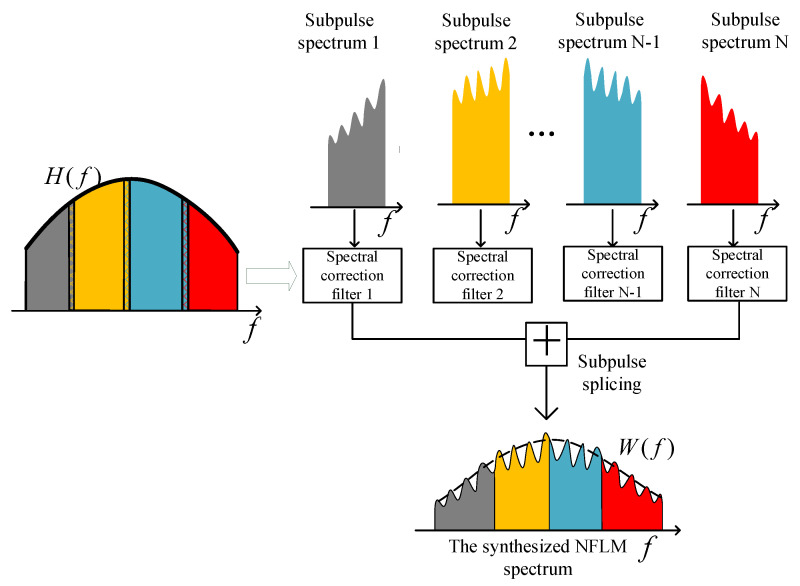
The optimization of subpulse matched filters.

**Figure 8 sensors-22-09829-f008:**
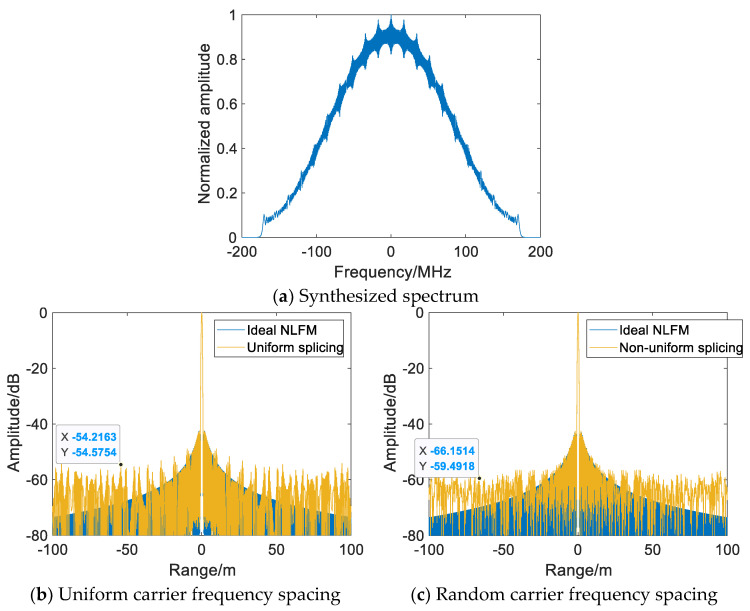
Range-compressed NLFM SWW after spectral correction *N* = 20 *B* = 20.7 MHz Δ*f* = 17.2 MHz *T_syn_* = 145 µs.

**Figure 9 sensors-22-09829-f009:**
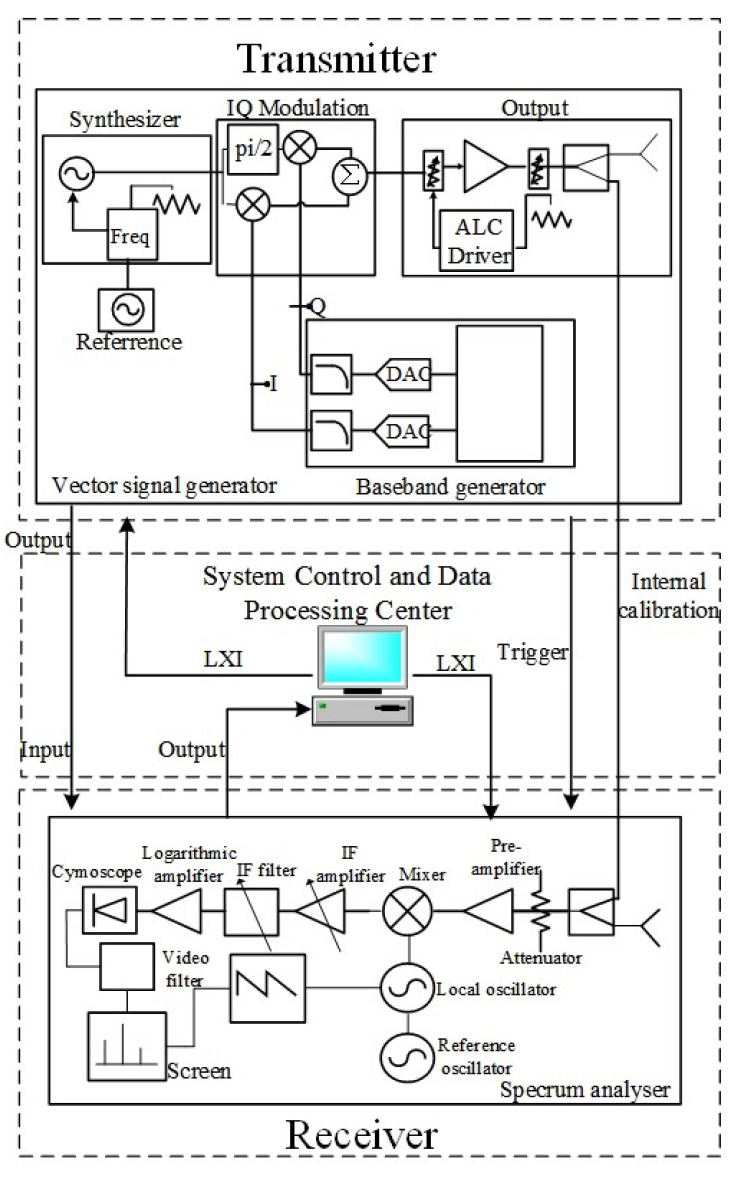
The connection block diagram.

**Figure 10 sensors-22-09829-f010:**
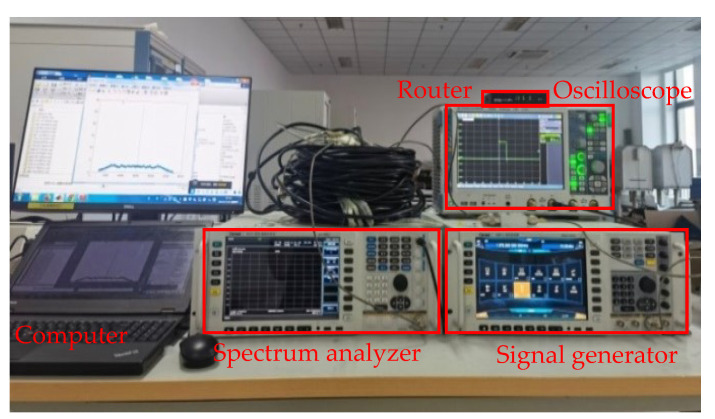
Hardware-in-the-loop simulation system for SF FM radar.

**Figure 11 sensors-22-09829-f011:**
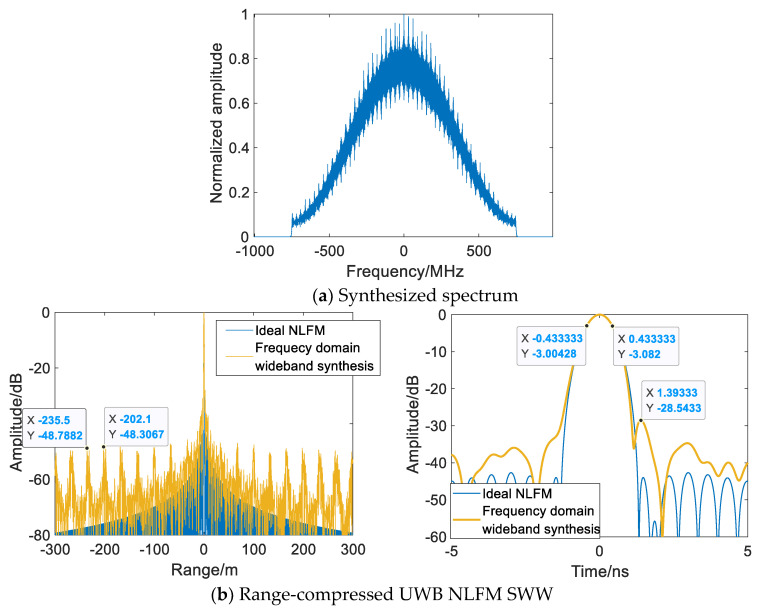
Frequency domain wideband synthesis method.

**Figure 12 sensors-22-09829-f012:**
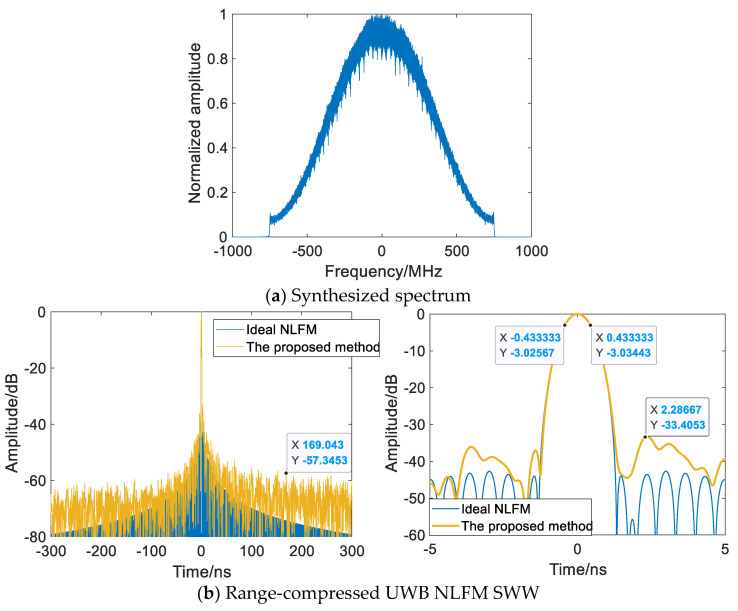
Multisubpulse UWB NLFM waveform synthesis method.

**Figure 13 sensors-22-09829-f013:**
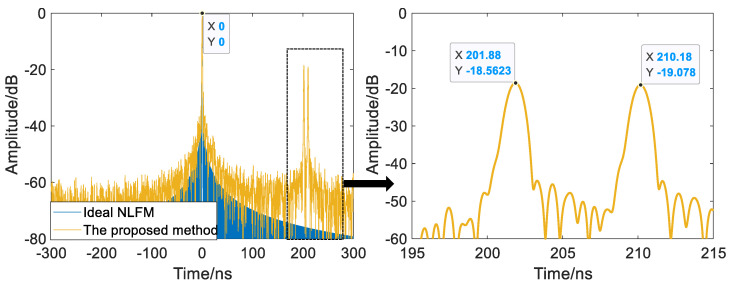
The results of the two-point experiment.

**Figure 14 sensors-22-09829-f014:**
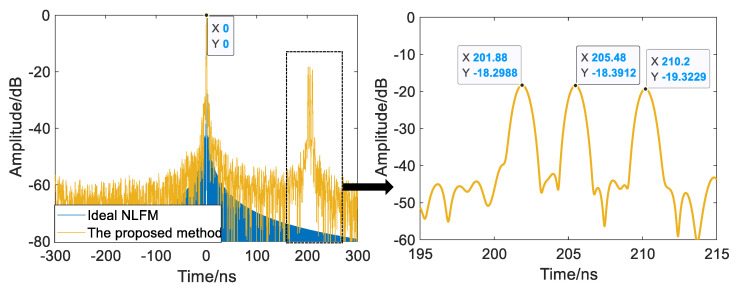
The results of the three-point experiment.

**Table 1 sensors-22-09829-t001:** Performance indicator of range-compressed NLFM SWW.

	Bandwidth/MHz	Range Resolution/m	ISLR/dB	PSLR/dB
Ideal NLFM waveform	347.5	0.5615	−34.2	−42.7
Uniform carrier frequency spacing	347.5	0.5615	−22.4	−41.2
Nonuniform carrier frequency spacing	347.5	0.5615	−22.6	−42.1
Uniform carrier frequency spacing (spectral correction)	347.5	0.5615	−27.2	−42.2
Nonuniform carrier frequency spacing (spectral correction)	347.5	0.5615	−27.5	−42.5

**Table 2 sensors-22-09829-t002:** Experimental parameters.

Parameter	Value
Start frequency/MHz	508
Stop frequency/MHz	1978
Subpulse bandwidth/MHz	36
Reference frequency/MHz	10
Frequency step size/MHz	30
Sampling rate/MHz	50
Output power/dBm	10
Number of subpulses	50
Window	Hamming

**Table 3 sensors-22-09829-t003:** Performance indicator of range-compressed UWB NLFM SWW.

	Bandwidth/MHz	Time Resolution/ns	ISLR/dB	PSLR/dB
Ideal NLFM waveform	1506	0.8667	−34.3	−43.9
Frequency domain wideband synthesis	1506	0.8667	−19.3	−28.6
The proposed method	1506	0.8667	−23.8	−33.4

**Table 4 sensors-22-09829-t004:** The group delay of point targets.

Group Delay/ns	Two-Point Target Experiment	Three-Point Target Experiment
Target 1	201.914	201.914
Target 2	210.094	205.521
Target 3		210.094
Minimum delay difference between targets	8.180	3.607

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
