# Peer review of "Research on Ultra-Wideband NLFM Waveform Synthesis and Grating Lobe Suppression"

_sensors, 2022, doi:10.3390/s22249829_

Round 1

Reviewer 1 Report

Dear Authors,

Please, find attached some comments. In fact, you need to explain more about the method you used in experimental section.

Author Response

Please, find attached some comments. In fact, you need to explain more about the method you used in experimental section.

(1) The reference error in the article has been corrected.

(2) The formulas marked in the attached PDF have been corrected.

(3) Bibliographic citations in the Introduction section have been supplemented.

(4)Based on the SF FM radar hardware-in-the-loop simulation system, another paper of mine discusses in detail the problems and solutions of wideband waveform synthesis in practical radar applications; however, it cannot be included in the reference because it has not been published online. In the paper, system distortion, delay error, and phase jump at sub-pulse splicing that impact SWW performance in actual system applications are improved by pre-distortion, high-precision delay alignment, and phase adaptive adjustment technologies, and the method's efficacy is demonstrated via the hardware-in-the-loop simulation system. In addition, the experimental method is briefly supplemented in Section 4.1 of this paper.

Reviewer 2 Report

The submitted manuscript presents research on UWB NLFM waveform synthesis and grating lobe suppression. The research is interesting and few comments are: 

1-    The English of the manuscript should be improved.

2-    According to MDPI referencing style, the reference citation number must be placed in square brackets [].

3-    The number found in line 22 (abstract section) needs a correction.

4-     The introduction section is short. Add one or two more paragraphs to it, and summarize it in detail with recently published works (last two years).

5-    The author should rewrite all equations in the manuscript correctly. I suggest for the authors use the MS equation editor with MS word instead of using the MathType equation editor.

6-    I guess there's a misplaced dot after the caption of Figure 1.

7-    I guess there's a misplaced dot after the word “Figure” in line 277.

8-    A performance comparison table with recently published papers using other approaches is recommended.

Author Response

Dear reviewer:

Thank you for your decision and constructive comments on my manuscript. We have carefully considered the suggestion of Reviewer and make some changes. We have tried our best to improve and made some changes in the manuscript. Revision notes, point-to-point, are given as follows:

  • The English of the manuscript should be improved.

Response:This article has been professionally refined to correct most grammatical errors and inaccuracies.

  • According to MDPI referencing style, the reference citation number must be placed in square brackets [].

Response:The reference error in the article has been corrected.

  • The number found in line 22 (abstract section) needs a correction.

Response:The ambiguous 50 36 MHz sub-pulse in the abstract has been changed to 50 sub-pulses with a bandwidth of 36 MHz.

  • The introduction section is short. Add one or two more paragraphs to it, and summarize it in detail with recently published works (last two years).

Response:The relevant research in the past two years has been supplemented, and the introduction part has been added. The supplementary content is as follows: By developing an accurate and approximative model of the LFM sub-pulse spectrum with a small TxBW, Reference [5] proposes a low blind distance ultra-wideband wave-form design method based on spectral fluctuation period and Fresnel integral win-dowing that reduces the number of grating lobes by over fifty percent and suppresses the highest grating lobe level by at least 4 dB. However, the sub-pulse spectrum in NLFM SWW is distinct, and the derivation of the Fresnel integral is complex, making it challenging to apply to NLFM SWW. Reference [6] determines that intra-pulse and in-ter-pulse amplitude and phase errors introduced by device non-ideality are the cause of grating lobes and proposes using strong scattering point echoes for amplitude and phase error compensation, as well as providing a grating lobe suppression method of LFM SWW from a system application perspective.

  • The author should rewrite all equations in the manuscript correctly. I suggest for the authors use the MS equation editor with MS word instead of using the MathType equation editor.

Response:All formulas have been rechecked and rewritten in MS Word.

6-    I guess there's a misplaced dot after the caption of Figure 1.

Response: The error in the caption of Figure 1 has been corrected

7-    I guess there's a misplaced dot after the word “Figure” in line 277.

Response: The error in line 277 has been corrected

8-    A performance comparison table with recently published papers using other approaches is recommended.

  • The main work of this paper is to extend the frequency domain wideband synthesis method of LFM waveform to ultra-wideband NLFM waveform synthesis. The majority of existing wideband synthesis methods are for LFM SWW. Wideband synthesis methods of LFM waveforms cannot be directly applied to NLFM SWW because the time-frequency variation law of the signal and the Fresnel ripple characteristics of the sub-pulse spectrum are dissimilar. For example, Reference 5 is based on the spectral fluctuation period of the LFM sub-pulse and the Fresnel integral window to suppress grating lobes of the LFM SWW; however, in the NLFM SWW, the spectral fluctuation period of the sub-pulses is not equal, and the Fresnel integrals of each sub-pulse are distinct and difficult to deduce. Consequently, the further research will continue to enhance these algorithms for the characteristics of NLFM waveforms and compare their performance.
  • The work on wideband synthesis of NLFM waveforms is still in its early stages. The SF FM sub-pulse model is built on the characteristics of NLFM waveforms in this paper, and the frequency domain wideband synthesis of LFM waveforms is applied to the synthesis of NLFM waveforms. In order to suppress high grating lobes caused by equidistant splicing and small TxBW, corresponding improvement methods based on random frequency hopping and spectrum correction technology are proposed. Other factors, however, will have an impact on the performance of NLFM SWW. In the future, we will continue to study the grating lobe suppression of NLFM SWW and optimize the NFLM waveform synthesis method.

Reviewer 3 Report

This manuscript presents a study on the generation of ultra-wideband nonlinear frequency modulation signals combined with grating lobe suppression.
A hardware-in-the loop testbed is used in the studies to validate the theory.
The main claim is an improvement of at least 4dB and I think that the numbers must be rounded to the tenths of dB at most.
Moreover, I think that hundreds of micro-dB is not accepted on a journal paper, especially for these transactions.
The explanation of these results must be improved in the text:
(1) Random carrier frequency spacnig: this procedure reduces the grating lobes (Figura 4d)
(2) Multi-sub-pulse UWB NFLM waveform synthesis method improves the spectrum (Figure 8d)
The setup must be detailed in section 4.1 to allow the reader to replicate the results.
Table 3: how did the ISLR/dB and PSLR/dB were obtained?
The last remark is how this work relates to a sensors applications other than radar sensors in general and 2D imaging algorithms and SF FM radar in particular.

Author Response

Dear reviewer:

Thank you for your decision and constructive comments on my manuscript. We have carefully considered the suggestion of Reviewer and make some changes. We have tried our best to improve and made some changes in the manuscript. Revision notes, point-to-point, are given as follows:

1.The main claim is an improvement of at least 4dB and I think that the numbers must be rounded to the tenths of dB at most.Moreover, I think that hundreds of micro-dB is not accepted on a journal paper, especially for these transactions.

Response:I only gave the corresponding results from the perspective of numerical calculation when using PSLR and ISLR to measure the performance of range-compressed SWW. In practical applications, PSLR and ISLR are approximately 0.1dB enough to meet the requirements. Indicators used to measure the performance of the range-compressed SWW, such as ISLR and PSLR, are approximated to 0.1dB

  1. The explanation of these results must be improved in the text:

(1) Random carrier frequency spacing: this procedure reduces the grating lobes (Figure 4d)

Response: The reason why the grating lobe can be effectively suppressed by non-uniform splicing in Figure4(d) is supplemented above Figure4. The content is as follows: Figure 4 shows the range-compressed NLFM SWW synthesized by 8 sub-pulses based on the Hamming window. There are periodic grating lobes in range at uniform carrier frequency spacing; the difference between the time-bandwidth products of the sub-pulses is small, and the sub-pulse spectra on both sides are less impacted by the time-bandwidth products [18]. At this time, grating lobes are mainly caused by equally spaced splicing, so most of the grating lobes can be suppressed at non-uniform carrier frequency spacing.

(2) Multi-sub-pulse UWB NFLM waveform synthesis method improves the spectrum (Figure 8d)

Multi-sub-pulse UWB NFLM waveform synthesis method combines the advantages of spectral correction and random hooping frequency technology by optimizing the sub-pulse matched filter through spectral correction to suppress spectrum edge jump and in-band ripple of the sub-pulse, thereby reducing the Fresnel ripple and gap caused by the small time bandwidth product in the synthesized spectrum, and dispersing the energy of periodic grating lobes via non-uniform splicing. The comparison of Figures 8c, 8b, and 5 indicates that the PSLR and ISLR of NLFM SWW by the proposed method integrating the above two technologies are superior to the traditional method and uniform splicing, demonstrating the proposed method's effectiveness. A supplementary note has been added above Figure 8c (I think you mean 8(c)), which reads: In conclusion, the multi-sub-pulse UWB NFLM waveform synthesis method can dis-perse the periodic grating lobe energy and suppress the grating lobe by reducing in-band ripples and notches of the synthesized spectrum and breaking the periodicity of the discontinuity. Both the PSLR and ISLR of SWW are superior to those of the traditional wideband synthesis method and uniform splicing, proving the effectiveness of the proposed method.

  1. The setup must be detailed in section 4.1 to allow the reader to replicate the results.

Response: Based on the SF FM radar hardware-in-the-loop simulation system, another paper of mine discusses in detail the problems and solutions of wideband waveform synthesis in practical radar applications; however, it cannot be included in the References because it has not been published online. In the paper, system distortion, delay error, and phase jump at sub-pulse splicing that impact SWW performance in actual system applications are improved by pre-distortion, high-precision delay alignment, and phase adaptive adjustment technologies, and the method's efficacy is demonstrated via the hardware-in-the-loop simulation system. In addition, the experimental method is briefly supplemented in Section 4.1 of this paper.

  1. Table 3: how did the ISLR/dB and PSLR/dB were obtained?

Response:The peak side lobe ratio(PSLR) is the ratio of the peak intensity of the side lobe to the peak intensity of the strongest main lobe. The integral side lobe ratio is the ratio of the total energy of the side lobes to the total energy of the main lobes .The ISLR and PSLR are calculated according to their definitions, respectively, and the formulas are as follows:

  1. The last remark is how this work relates to a sensors applications other than radar sensors in general and 2D imaging algorithms and SF FM radar in particular

Response:(1) Currently, we are also conducting research on the SF FM real-time radar system, which requires the radar system's transmitting unit to have an arbitrary waveform generator and the receiving unit to be capable of dual-channel data acquisition. One channel is used to receive the internal calibration signal, while the other is utilized to receive the target's echo.

(2)A limitation of frequency NLFM SWW is their susceptibility to the

decoherence effects caused by platform motion. When NLFM SWW is utilized for 2D imaging, ultra-wideband synthesis processing must be performed on each sub-pulse in the range direction prior to pulse compression in the azimuth direction. Compared to conventional 2D imaging algorithms, the correlation between azimuth signals will decrease, and the motion compensation caused by a moving platform will become more complex. Therefore, it will be essential in the future to conduct research on 2D imaging algorithms for NLFM SWW, to perform motion compensation based on the platform's motion characteristics, and to generate azimuth-matched filters in order to acquire high-quality radar images.

Round 2

Reviewer 1 Report

Thank you for improving the manuscript.

Reviewer 2 Report

The authors respond to all comments. The manuscript can be accepted in its current form.